# Al–Al_3_Ni In Situ Composite Formation by Wire-Feed Electron-Beam Additive Manufacturing

**DOI:** 10.3390/ma16114157

**Published:** 2023-06-02

**Authors:** Artem Dobrovolskii, Andrey Chumaevskii, Anna Zykova, Nikolay Savchenko, Denis Gurianov, Aleksandra Nikolaeva, Natalia Semenchuk, Sergey Nikonov, Pavel Sokolov, Valery Rubtsov, Evgeny Kolubaev

**Affiliations:** Institute of Strength Physics and Materials Science, Siberian Branch of Russian Academy of Sciences, 634055 Tomsk, Russia; artdobrov@ispms.ru (A.D.); zykovaap@mail.ru (A.Z.); savnick@ispms.tsc.ru (N.S.); desa-93@mail.ru (D.G.); philip371g@gmail.com (A.N.); natali.t.v@ispms.ru (N.S.); sergrff@ngs.ru (S.N.); sps11@sibmail.com (P.S.); rvy@ispms.ru (V.R.); eak@ispms.ru (E.K.)

**Keywords:** wire feed electron beam additive manufacturing, electron beam freeform fabrication, aluminum alloys, nickel superalloys, multiphase materials, in situ composites

## Abstract

The regularities of microstructure formation in samples of multiphase composites obtained by additive electron beam manufacturing on the basis of aluminum alloy ER4043 and nickel superalloy Udimet-500 have been studied. The results of the structure study show that a multicomponent structure is formed in the samples with the presence of Cr_23_C_6_ carbides, solid solutions based on aluminum -Al or silicon -Si, eutectics along the boundaries of dendrites, intermetallic phases Al_3_Ni, AlNi_3_, Al_7_5Co_22_Ni_3_, and Al_5_Co, as well as carbides of complex composition AlCCr, Al_8_SiC_7_, of a different morphology. The formation of a number of intermetallic phases present in local areas of the samples was also distinguished. A large amount of solid phases leads to the formation of a material with high hardness and low ductility. The fracture of composite specimens under tension and compression is brittle, without revealing the stage of plastic flow. Tensile strength values are significantly reduced from the initial 142–164 MPa to 55–123 MPa. In compression, the tensile strength values increase to 490–570 MPa and 905–1200 MPa with the introduction of 5% and 10% nickel superalloy, respectively. An increase in the hardness and compressive strength of the surface layers results in an increase in the wear resistance of the specimens and a decrease in the coefficient of friction.

## 1. Introduction

Cast Al-Si alloys are commonly used in automotive engines due to their good castability, high strength-to-weight ratio, excellent corrosion resistance, and low expansion ratio coefficient [1,2]. Aluminum-silicon alloys (silumins) are used to produce body parts, shaped castings, and drawn and welding wires [1,2]. The negative characteristics of silumins are low tensile strength (135–235 MPa) and the tendency to brittle fracture (relative elongation less than 4%) [3,4]. Heat treatment, work hardening, and alloying can be used to improve the properties of aluminum alloys [5,6]. Ni, Ti, Mo, Cu, Mg, and other elements are used as alloying components [7,8,9,10,11,12]. To improve low mechanical properties, the fabrication of composite materials can be considered.

The production of composite materials with a metal matrix on the basis of aluminum alloys has wide application prospects, since it allows us to obtain products with a hardened structure of individual components of the product, while retaining the properties of the base metal with low cost and density, as well as high plasticity [13,14,15,16,17]. The use of modern production methods makes it possible to produce lightweight nickel-modified aluminum products that are resistant to frictional wear due to hardening with Al_3_Ni intermetallic particles [18,19,20,21,22,23,24,25,26,27,28,29,30,31,32,33]. The bulk intermetallic phase of Al_3_Ni exhibits an orthorhombic crystal structure and has a density of 4000 kg/m^3^ and an elastic modulus of 140 MPa [34].

The intermetallic phase Al_3_Ni is formed in alloys with a high content of Al alloyed with Ni. This solid phase is a promising reinforcing material for improving the properties of aluminum alloys at high temperatures. The brittle nature of the Al_3_Ni intermetallic phase limits its use unless it is uniformly dispersed in softer metal matrices. The primary Al_3_Ni intermetallic phase is observed in Al–Ni alloys containing more than 5 wt. % Ni [35], while, conversely, the AlNi intermetallic phase becomes the main phase at an Ni content of 45 wt. % and higher [36].

Several processes for manufacturing Al_3_Ni–Al composites exist, such as mechanical alloying [28], equal-channel angular pressing [29], directional solidification [30], the electromagnetic separation method [31], and friction stir processing [32,33].

One of the modern methods of manufacturing composite materials is two-wire electron beam additive manufacturing using wire filaments. This method is a type of electron beam additive manufacturing. The essence of the method is the sequential deposition of wire material layer by layer on a cooled substrate. The filament deposition is performed by electron beam melting in a vacuum chamber, while the filament is fed into the melting zone by wire feeders [37,38,39,40,41]. A feature of the process is the simultaneous feeding of two wires into the melt zone, resulting in the local alloying of a model product with the formation of a more complex multiphase structure [42,43]. This method is of great interest to the scientific community because it allows the production of a wide variety of materials based on standard metal wires by using two wire feeders with controlled filament feed into the melt bath. As a result, both the formation of composite materials with a non-standard combination of structure and properties and the production of parts with a directional structure in various areas are possible [44,45].

At present, although there are a number of publications in the literature on the preparation of composite and functionally graded materials by wire electron beam technology, there is only a small amount of information on the preparation of multicomponent composite materials, especially those based on light aluminum or titanium alloys [46,47,48]. In this paper, we considered the formation of the structure of multiphase composites based on the aluminum-silicon alloy ER4043 (AlSi_5_) and the nickel superalloy Udimet-500 and its influence on the mechanical properties of the obtained materials. Such a combination of the base metal and the hardening additive will allow the study of the patterns of material formation during the interaction of the base metals of the alloys and their alloying elements. From the point of view of practical application, the preparation of these materials has prospects in the manufacture of parts based on aluminum alloys with surface layers of high hardness and wear resistance, as well as a strong and ductile base.

## 2. Materials and Methods

The samples were fabricated on a three-axis multi-beam electron beam additive manufacturing tool in a vacuum environment on a cooled AlMg_5_ aluminum alloy substrate. Model products were fabricated in the form of thin walls 8 mm thick, 80 mm high, and 120 mm long by linearly moving the coordinate table along one of the axes, layer by layer. The electron gun and wire feeders were static. ER4043 (“MetPromStar”, Moscow, Russia) and Udimet-500 (ESAB, North Bethesda, MD, USA) welding wires with a diameter of 1.2 mm were used as filaments. A Niton XL3t 980 GOLDD (Thermo Fisher Scientific, Waltham, MA, USA) X-ray fluorescence analyzer was used to control the chemical composition of the wires and the resulting model products (Table 1). In addition to these components, the Udimet-500 alloy contains up to 0.1% carbon.

Printing was performed according to the scheme shown in Figure 1. Pattern 1 was formed on the surface of the AA5056 alloy substrate 2 by feeding filaments 4 through nozzles 3 into the printing zone. The filaments 4 were melted by an electron beam 5 supplied from an electron gun 6 through a magnetic focusing system 7. As a result, a melt bath 8 was formed in the printing zone.

The pattern was formed layer by layer. The wire feed was not controlled during the printing process. Initially, the feed was adjusted based on achieving a volume concentration of the injected nickel alloy of 5 and 10 vol% of the composite volume. The comparison of the structure and properties of the composite materials was carried out with a pure AlSi5 alloy printed using the method of electron beam additive technology. This scheme was previously used for the preparation of multicomponent metal-matrix composites based on aluminum-manganese bronze with the introduction of nickel superalloys [48,49].

To reveal the structural elements, the samples were etched with Keller’s reagent (2.5 mL HNO_3_, 1.5 mL HCl, 1 mL HF, 95 mL H_2_O) after pregrinding and polishing. An Altami-MET 1C (Altami Ltd., Saint-Petersburg, Russia) optical microscope and an Olympus LEXT 4100 (Olympus NDT, Inc., Waltham, MA, USA) confocal microscope were used to examine the microstructure of the samples. Microscopic studies were carried out on an Apreo 2 S (Thermo Fisher Scientific, Waltham, MA, USA) scanning electron microscope. X-ray diffraction analysis was performed with an X-ray diffractometer XRD-7000S (Yekaterinburg, Russia), CoKα. Transmission microscopy was performed on a JEOL-2100 (JEOL Ltd., Akishima, Japan) universal transmission microscope. Mechanical tests were performed on a UTS-110M (Testsystems, Ivanovo, Russia) machine. The size of the compression specimens was 3 × 3 × 6 mm. The size of the working part of the tensile specimens was 12 × 2.5 × 2.5 mm. The strain rate during compression and tension was 1 mm/min. Tribological tests for dry friction according to the “pin on disk” scheme were carried out on a TRIBOTECHNIC (Tokyo Boeki Group, Japan, Tokyo) tribometer paired with a counter body in the form of a 5 mm diameter ball at sliding speeds of 0.06, 0.15, and 0.24 m/s, and a normal compressive force of 10 N. The size of samples for tribological tests was 3 × 3 × 10 mm. The rotational speed was 200 rpm and the test duration was 300 min. The change in the friction rate occurred due to the displacement of the mating body at a distance of 3, 7, and 12 mm from the center of the disk.

## 3. Results

In the presented optical images of the microstructure of the hypoeutectic aluminum-silicon alloy AlSi5 obtained using the EBAM method, aluminum-based α-solid solution dendrites, and fine-grained eutectics between the dendrites are observed (Figure 2). Such a structure is typical for aluminum-silicon alloys obtained by casting or electron beam printing [50]. The structure of the obtained material differs significantly in the upper (Figure 2a), middle (Figure 2b), and lower (Figure 2c) parts of the sample. In the upper part of the sample, the volume fraction of the eutectic between the α-Al dendrites increases and the size of the dendritic branches decreases.

The formation of the Al-Ni5% composite material leads, on the one hand, to the formation of various intermetallic and other phases in the structure of the samples and, on the other hand, to the formation of sufficiently large fragments of the Udimet-500 alloy, which only partially interacted with the aluminum matrix (Figure 3). The appearance of relatively large fragments of the nickel alloy, which only partially reacted with the matrix, is due to its high melting point compared to the AlSi5 alloy, as a result of which the material introduced into the melt bath crystallizes much earlier than the aluminum alloy and settles in the lower part of the layer, incompletely mixing with aluminum.

The size of the aluminum-based solid solution dendrites 1 in the composite is significantly smaller than in the unmodified material due to the presence of intermetallic compounds 3 and partially reacted nickel alloy 4 in the structure, which limit the growth of aluminum dendrites (Figure 2). The volume fraction of eutectic 2 is significantly reduced compared to the pure AlSi5 alloy (Figure 2 and Figure 3). As a result of the interaction of nickel alloy components with an aluminum matrix, many structural components of different chemical compositions with different particle morphologies from globular to acicular inclusions are formed. In the structure of composite materials, the presence of microcracks is observed as a result of thermal stresses during cooling of the material, differences in the coefficients of thermal expansion of the components and low plasticity of intermetallic phases.

As the nickel alloy concentration increases, the proportion of intermetallic inclusions increases and the size and volume fraction of α-solid solution dendrites in aluminum decreases (Figure 4). In the composite structure in Figure 4, practically no eutectics (α-Al+β-Si) are released. Intermetallic phases are formed from lamellar to almost equiaxed. The structure in the upper part of the samples is significantly enlarged compared to the lower part. Large fragments of a partially reacted nickel alloy are also present, consisting partly of intermetallic phases of different compositions and partly of nickel superalloy fragments.

The phase composition of the samples, determined by X-ray diffraction analysis, is identical for different concentrations of the components contained in the composites (Figure 5). In addition to the initial components of the aluminum alloy α-Al and β-Si, the structure of the composites contains intermetallic compounds Al_3_Ni, Al_75_Co_22_Ni_3_ and Al_5_Co, as well as carbides of complex composition AlCCr, Al_8_SiC_7_. The structure in the lower, upper, and middle parts of the specimens also does not differ practically in the phase composition.

The microstructure of AlSi5 alloy is characterized by dendritic cells of solid solution, along the boundaries of which the eutectic (α-Al+β-Si) is located (Figure 2 and Figure 6). This is confirmed by TEM data, according to which the reflections (111) and (002) contain -Si and -Al phases, respectively (spectrum 1–3 in Figure 7). According to EDS analysis, up to 1.5 at.% Fe also dissolves in the silicon particles (spectrum 2 and 3 in Figure 7a). The microstructure of the AlSi5 alloy is similar to that of a cast aluminum-silicon alloy. However, unlike the AlSi5 cast alloy obtained by the wire additive electron beam technique, the eutectic Si is not in the form of plates, but rather spherical particles of various sizes (Figure 6). In addition to fine silicon particles, the eutectic also contains rather large silicon particles, reaching a size of 3–5 µm (Figure 6).

In the area of uneven mixing of the components of 2 wires of the composite material, a chemical composition gradient is formed (Figure 8a,b). This leads to the formation of a complex multiphase structure. The regions of partially mixed components are a mixture of AlNi and Al_3_Ni intermetallic grains enriched in Cr (up to ~5 at.%), Co (~6 at.%), and Si (~5 at.%) (Figure 8a, spectra 1–3, Table 2). Moving away from such regions, the solid solution contains Al_3_Ni particles, in which the total concentration of Cr, Co, and Si does not exceed 7 at.% (Figure 8a, spectrum 6, Table 2). An insignificant number of impurities are observed in the α-Al solid solution (Figure 8a,d, spectra 5 and 7, Table 2). In general, in the composite where the components were completely mixed, the structure is characterized by a solid solution with eutectic (α+β), grains of supersaturated solid solution of Co, Ni, Si in -Al, and Al_3_Ni particles (Figure 8d–i).

The SEM-EDS data compare well with the TEM-EDS data (Figure 9 and Table 3). Particles of lamellar (Figure 9a–f) or spherical (Figure 9g–i) shape are also formed. The composition of particles of similar shape is identical. The major part of the material is occupied by solid solutions of α-Al and β-Si (3, 4, 5, 6, 9 at. Figure 9 and Table 3). Intermetallics of the complex composition Al_3_(Ni,Cr,Si) and Al(Co,Ni) (1, 2, 7, 8 at. Figure 9 and Table 3) are also distinguished in the structure. This is due to the presence of chromium and cobalt in the Udimet-500 alloy and silicon in the aluminum alloy, resulting in the formation of intermetallic compounds of a more complex composition than Al_3_Ni or AlNi. The intermetallic phases are predominantly lamellar or irregular, while the silicon particles are spherical (Figure 9).

In addition to the formation of the particles of the above intermetallic compounds, intermetallic phases based on aluminum and cobalt are formed (Figure 10). According to the results of transmission microscopy, Al_9_Co_2_ particles are formed in the samples (Figure 10). The chemical analysis data of individual sections also show the formation of Al_9_Co_2_Ni. In this case, X-ray diffraction analysis shows the formation of intermetallic compounds of the complex composition, Al_75_Co_22_Ni_3_. In this case, it is rather difficult to determine which intermetallic phases are formed. Presumably, due to the inhomogeneity of the distribution of chemical elements, a fairly wide range of intermetallic compounds is formed in local areas of the sample, with the predominance of Al_75_Co_22_Ni_3_ in the main volume as a whole, which is shown by X-ray diffraction analysis.

Increasing the content of nickel alloy in the matrix of the composite up to 10% leads to the formation of a structure similar to that described above. However, the proportion of intermetallic phases increases and the content of the α-Al solid solution decreases (Figure 11 and Table 4). In those areas where poor mixing of the components is observed, chromium enrichment is observed along the grain boundaries of AlNi (1, 2 in Figure 11 and Table 4). Presumably, AlCCr carbide is formed (3 in Figure 11b and Table 4). AlNi intermetallic phases based on aluminum and nickel also contain some dissolved chromium and cobalt.

Therefore, in this case, the formation of AlNi(Cr,Co) and Al_3_(Ni,Cr,Si) intermetallic compounds of complex composition also takes place (1, 2, 4, 5, 6, 8, 11, 12 at Figure 11 and Table 4). Al_3_Ni grains with a number of impurities are also formed. The formation of Al_3_Ti intermetallic phases in the form of thin plates or needles is also observed (Figure 11q). The microstructure of the aluminum matrix composite and globular particles also contains components of the original aluminum-silicon alloy (Figure 12). There is a solid solution of α-Al with dissolved Si (Figure 12, item 2). Microdiffraction images obtained from particles (1, 4, 5 in Figure 12a,c) indicate that they are silicon particles (Figure 12b,d). However, EDS analysis of a particle (item 3 in Figure 12a) containing 75.2 at.% Al and 22 at.% Ni indicates that it is an Al_3_Ni particle. The particle also contains 1.1 at.% Si and 1.6 at.% Co.

As shown by SEM, the composite structure contains lamellar particles enriched with titanium, mainly located in the eutectic (α+β) of the α-Al matrix (Figure 13a,b). This is also confirmed by TEM results with electron microdiffraction interpretation (Figure 13a,c). These particles are Al_3_Ti; according to the EDS data, they also contain 6.9 at.% Si, 1.8 at.% Cr, and 2.7 at.% Mo. According to TEM data, the particle adjacent to Al_3_Ti is Al_3_Ni (Figure 13a,d).

The results of the uniaxial tensile tests show the brittle nature of the fracture of Al-Ni5% and Al-Ni10% composites caused by the formation of large intermetallic particles. Microcracking also occurs during the cooling of components with different crystallization temperatures and coefficients of thermal expansion. Cracks are formed mainly in intermetallic particles. When the additively produced AlSi5 aluminum alloy is modified with a nickel alloy during printing, the ultimate strength decreases from the initial 142–164 MPa to 55–123 MPa (Table 5 and Figure 14). Since the volume fraction of intermetallic phases is at a maximum in the Al-10%Ni composite, the least plastic and strong material is formed in this case.

Uniaxial compression tests show a significant increase in tensile strength due to multiphase hardening. Al-Ni5% composites have a tensile strength of 490–500 MPa in the horizontal direction and 530–570 MPa in the vertical loading direction (Figure 14 and Table 5). According to the results obtained, the tensile strength of Al-Ni10% composites is 905–945 MPa in the horizontal direction and 1150–1200 MPa in the vertical direction (Figure 14 and Table 5). The deformation behavior of the Al-5%Ni composite specimens is closest to that of the pure aluminum alloy specimens. At the same time, specimens of pure AlSi5 alloy practically do not break under compression and, although high plasticity is typical for Al-5%Ni specimens, at a strain higher than 0.075, the specimens break due to the formation of diagonal cracks. For Al-10%Ni specimens, due to the higher volume fraction of intermetallic compounds, higher strength is characteristic but fracture occurs without a pronounced plastic flow phase when the yield point is reached.

When the AlSi5 alloy is tested for wear resistance, no dependence of the coefficient of friction on the wear rate is observed (Figure 14e,f). The friction coefficient of the Al-5%Ni composite has a constant value of the friction coefficient of the order of 0.22–0.25 and does not depend on the sliding speed. The Al-10%Ni composite shows a decrease in the coefficient of friction from 0.46 to 0.36 with an increase in the friction speed from 0.06 to 0.24 m/s. The amount of wear, determined by the analysis of the cross profile of the friction marks, generally shows an increase with increasing friction velocity for all specimens (Figure 14f). The highest wear is typical for pure aluminum alloy samples. Average values are typical for Al-5%Ni samples. The minimum wear value for Al-10%Ni samples is when the effect of the friction speed on the degree of wear of the material is also the smallest.

## 4. Discussion

Al_3_Ni phase is the most favorable reaction in the aluminum-rich nickel region of the -Ni phase diagram [21]. The Gibbs free energy formations ΔGi (703 K) (kJ mol ^−1^) calculated by Qian et al. [32] connected with more low free energy at the Al_3_Ni particles: Ni_3_Al (Ni_0.75_Al_0.25_)—35.86 kJ mol^−1^; NiAl (Ni_0.50_Al_0.50_)—66.92 kJ mol^−1^; Ni_2_Al_3_(Ni_0.40_Al_0.60_)—61.94 kJ mol^−1^; NiAl_3_(Ni_0.25_Al_0.75_)—39.84 kJ mol^−1^.

The existence of Al_3_Ni XRD peaks (Figure 5) confirms that the intermetallic particles were formed due to the reaction between aluminum and nickel according to the equation 3Al + Ni → Al_3_Ni. The height of the XRD peaks of Al_3_Ni increases as the weight percentage of the nickel alloy increases. No nickel peaks were detected, suggesting that nickel was completely consumed in the reaction to form the intermetallic compound. This does not mean that the reaction time and holding period were sufficient for the complete synthesis of Al_3_Ni, since in the structure of the composite we find areas with an aluminum and nickel ratio corresponding to the AlNi compound (Figure 8 and Figure 9, Table 2 and Table 3). The fact that we do not see the AlNi phase using XRD (Figure 5) means that its relative amount is not large compared to Al_3_Ni.

As mentioned in the description of the results, the presence of Al_3_Ni particles correlated with a decrease in the size of the dendrites of the α-solid solution in aluminum. The reduction in aluminum grain size indicates that the Al_3_Ni particles act as a grain refiner. By all visibility synthesis, Al_3_Ni particles changed the picture solidification. A nucleus is necessary for birthing new grains during solidification. The growth of α -grains of aluminum is unhindered until the grain boundary meets the boundary of another growing grain. The synthesis of Al_3_Ni particles provides several grain nucleation regions for the production of new α-aluminum grains. The melting temperature of Al_3_Ni is higher than that of the aluminum matrix. The regions surrounding the Al_3_Ni particles are subjected to supercooling during solidification, which initiates the solidification of the aluminum grains. Growing aluminum grains encounter resistance to free growth due to the presence of Al_3_Ni particles in the aluminum melt. Increasing the content of Al_3_Ni particles increases the hindrance of free growth and increases the number of seed spots for α -aluminum grains.

The presence of pores in Figure 3 and Figure 4 may be due to the accumulation of brittle intermetallic particles, which are partially chipped during the preparation of thin sections to study the microstructure. Since EBAM is a multi-layer hardfacing process, residual stress buildup increases with the height of the hardfacing, which exacerbates the initiation of cracks in intermetallic grains.

Observed micrograph features, such as particle clumps, pores, and sharp particle corners, do not improve tensile behavior, while compressive strength and wear resistance levels improve with increasing amounts of synthesized intermetallic compounds (Figure 14).

## 5. Conclusions

The conducted studies show that it is possible to obtain multicomponent multimetal composite materials based on aluminum alloy ER4043 and nickel alloy Udimet-500 using the method of wire additive electron beam technology. The structure resulting from the interaction of the main and alloying elements is represented by a wide range of phases of complex composition and different morphologies. The main drawbacks of the obtained materials are the presence of inhomogeneities of various scale levels and the formation of microcracks in large particles of intermetallic compounds. The reasons for the formation of such defects are significant differences in the density and temperature of nickel and aluminum alloys, the formation of thermal stresses during cooling, and differences in the values of the coefficients of thermal expansion of the intermetallic phases and the metal matrix.

The results of the study of the structural phase state show that a multicomponent structure is formed in the samples with the presence of solutions based on aluminum -Al or silicon β-Si, intermetallic phases Al_3_Ni, Al_3_Ni, Al_75_Co_22_Ni_3_, and Al_5_Co, as well as carbides of complex composition AlCCr, Al_8_SiC_7_, which is confirmed by the results of both the energy dispersive analysis and the X-ray diffraction method. In addition to these phases, intermetallic compounds and solid solutions of AlNi, Al_3_Ti, Al_9_Co_2_, Al(Co,Ni), Al(Co,Ni,Si), Al_3_(Ni,Cr,Si), and AlNi(Cr,Co) are also found, which form in smaller amounts and are not identified by the X-ray diffraction method.

A large amount of solid phases leads to the formation of a material with high hardness and low ductility. The fracture of composite specimens under tension and compression is brittle without revealing the stage of plastic flow. Tensile strength values are significantly reduced from the initial 142–164 MPa to 55–123 MPa. In compression, the strength limits increase to 490–570 MPa and 905–1200 MPa with the introduction of 5% and 10% nickel superalloys, respectively. The amount of wear in dry friction tests decreases significantly in composite specimens of both compositions. Its dependence on the nickel concentration is rather ambiguous, although the maximum wear resistance is observed for specimens with a nickel alloy content of 10%. Thus, in addition to the basic possibility of obtaining multiphase composite materials of complex composition using the method of two-wire electron beam technology, it is also possible to obtain relatively wear-resistant products based on aluminum alloys.

## Figures and Tables

**Figure 1 materials-16-04157-f001:**
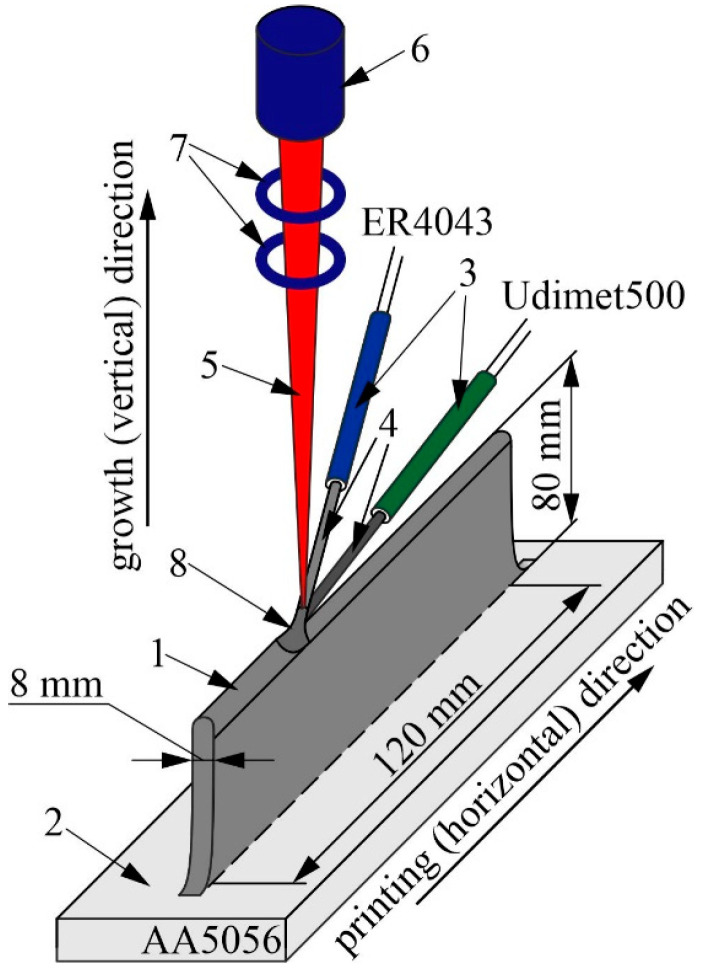
Schematic diagram of the printing process of multimetal composites based on AlSi5 aluminum alloy and Udimet-500 nickel superalloy using two-wire electron beam additive technology.

**Figure 2 materials-16-04157-f002:**
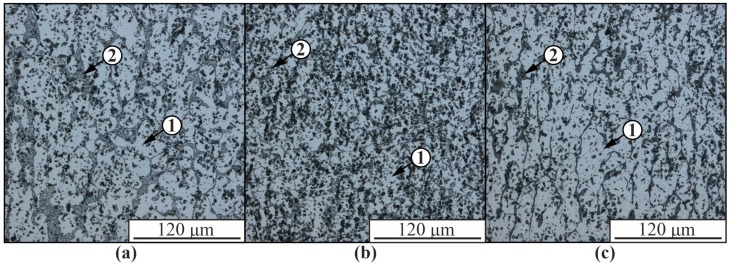
Optical images of the microstructure of ER4043 alloy obtained in the upper (**a**), middle (**b**), and lower (**c**) parts: 1—aluminum-based α-solid solution; 2—Al-Si eutectic.

**Figure 3 materials-16-04157-f003:**
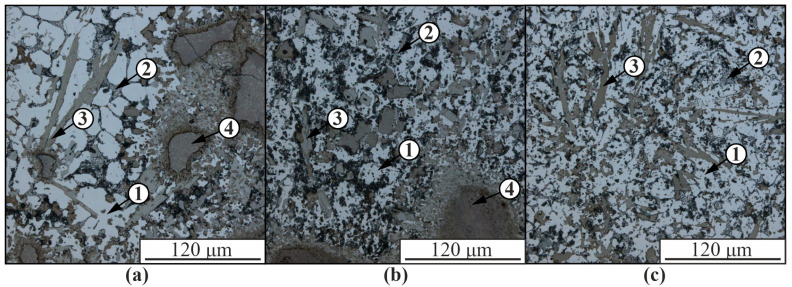
Optical images of the Al-Ni5% composite microstructure obtained in the upper (**a**), middle (**b**), and lower (**c**) parts: 1—aluminum-based α-solid solution; 2—Al-Si eutectic; 3—intermetallic phases; 4—a fragment of Udimet-500 nickel alloy partially reacted with the aluminum matrix.

**Figure 4 materials-16-04157-f004:**
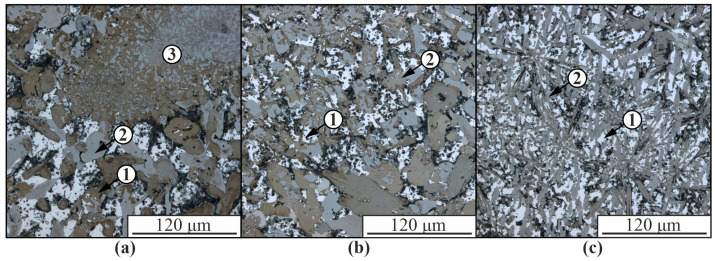
Optical images of the microstructure of the Al-Ni10% composite obtained in the upper (**a**), middle (**b**), and lower (**c**) parts. 1—Aluminum-based α-solid solution; 2—intermetallic phases; 3—a fragment of the nickel alloy Udimet-500 partially reacted with the aluminum matrix.

**Figure 5 materials-16-04157-f005:**
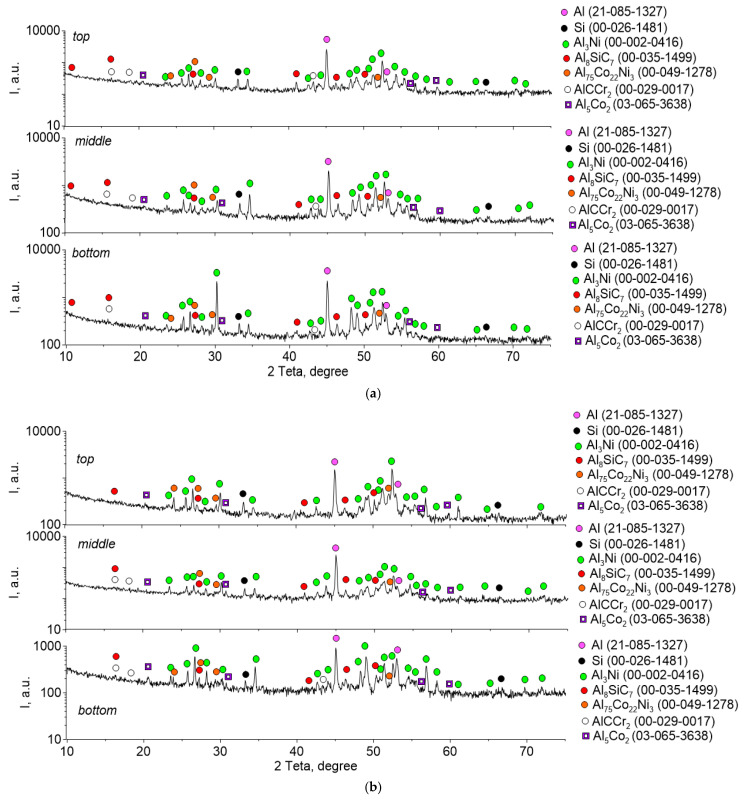
X-ray patterns of samples containing 5% (**a**) and 10% (**b**) nickel alloy Udimet-500 in an aluminum matrix.

**Figure 6 materials-16-04157-f006:**
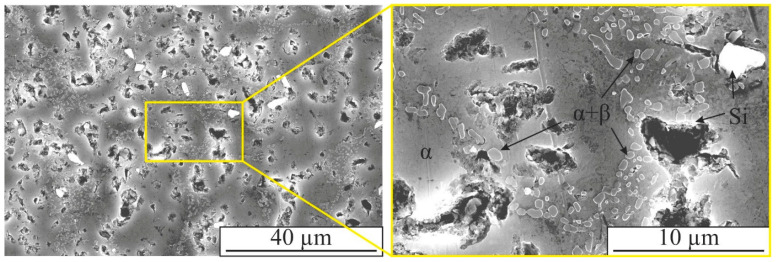
SE mode SEM images of the microstructure of the Al-8Si alloy obtained by EBAM.

**Figure 7 materials-16-04157-f007:**
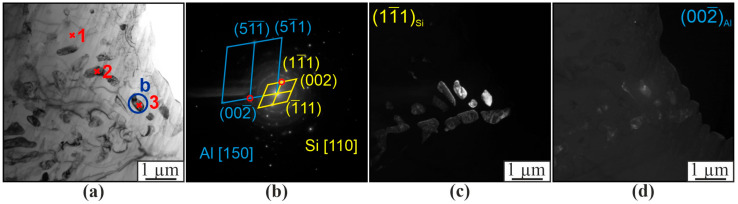
Bright-field TEM image of the microstructure of the ER4043 alloy (**a**), microdiffraction pattern (**b**) taken from a fragment of the area (**a**), dark-field images in the reflection (111)_Si_ (**c**) and in the reflection (002)_Al_ (**d**).

**Figure 8 materials-16-04157-f008:**
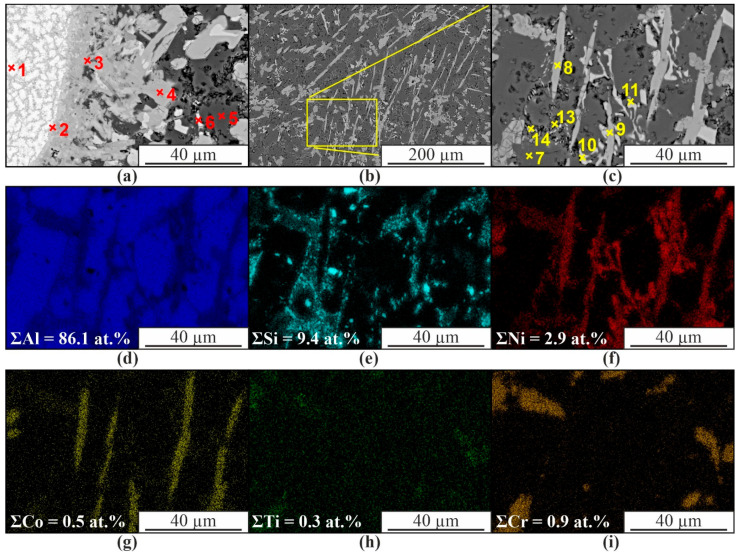
SEM images in BSE mode (**a**) and SE (**c**,**d**), EDS maps of element distribution (**e**–**i**) obtained from plot (**d**); (**b**) EDS analysis obtained along the line from plot (**a**).

**Figure 9 materials-16-04157-f009:**
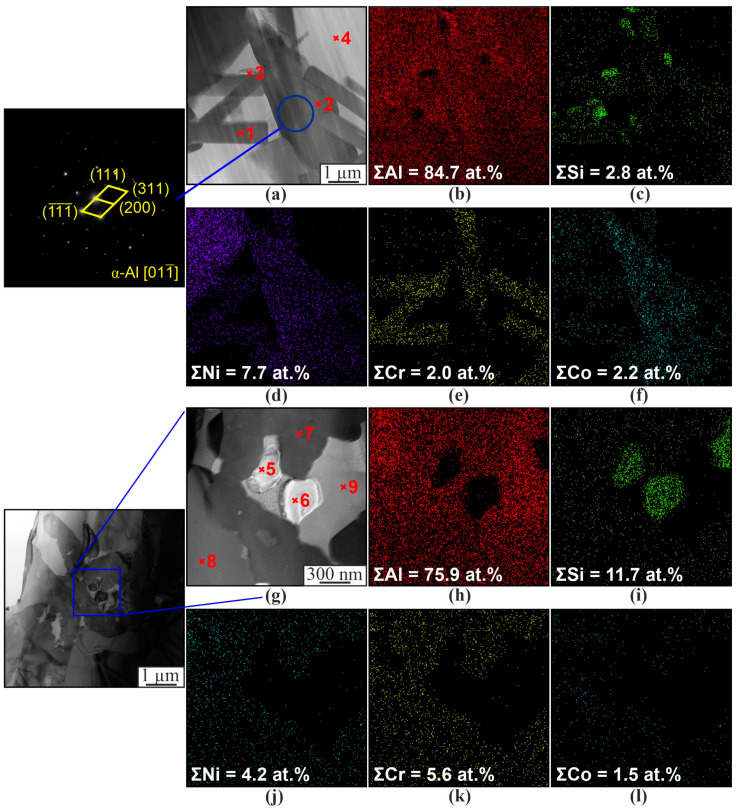
Microstructure of the composite material Al-5%Ni. Bright-field TEM image of the Al-5%Ni composite (**a**,**g**), and EDS element distribution maps (**b**–**f**,**h**–**l**).

**Figure 10 materials-16-04157-f010:**
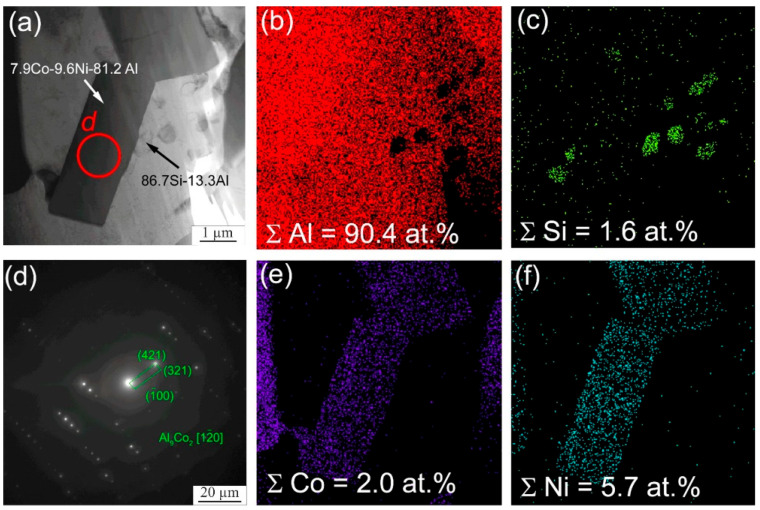
Bright-field TEM image of the intermetallic particles Al_9_Co_2_ (**a**), EDS element distribution maps (**b**,**c**,**e**,**f**) and SAED pattern from a particles Al_9_Co_2_ (**d**).

**Figure 11 materials-16-04157-f011:**
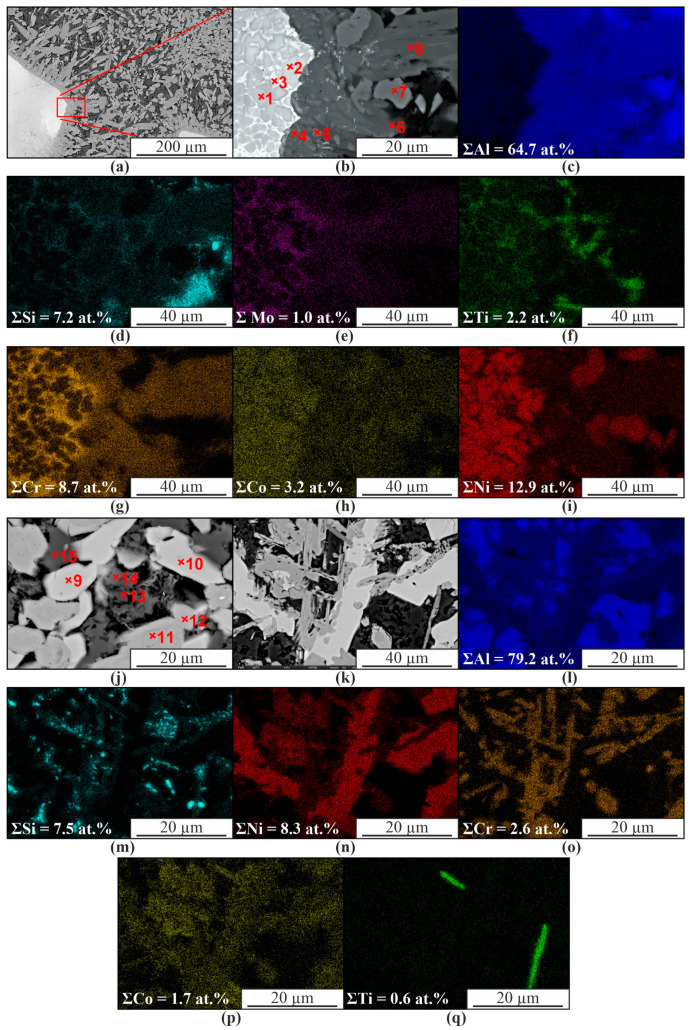
SEM images in BSE mode (**a**,**b**,**j**,**k**) and EDS maps of element distribution (**c**–**i**,**l**–**q**) obtained from the plot (**b**,**k**).

**Figure 12 materials-16-04157-f012:**
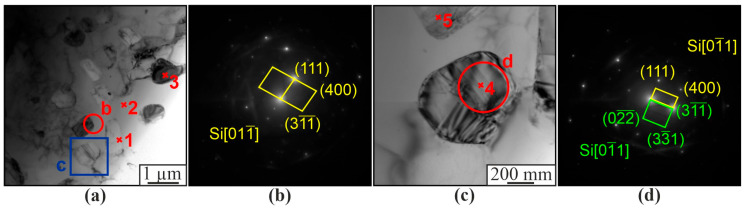
Bright field TEM images of a section of the microstructure of the Al-10% Ni composite (**a**,**c**), microdiffraction patterns (**b**,**d**) obtained from the area (**a**,**c**), respectively.

**Figure 13 materials-16-04157-f013:**
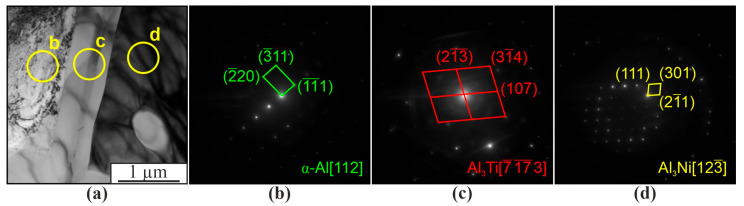
Bright field TEM image of a fragment of the microstructure of the Al-10% Ni composite (**a**) and microdiffraction patterns (**b**–**d**) obtained from the areas in figure (**a**).

**Figure 14 materials-16-04157-f014:**
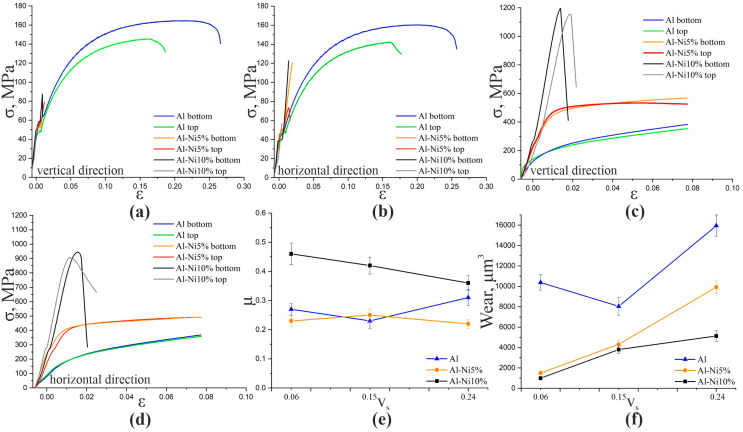
Engineering tensile stress–strain diagrams (**a**,**b**) and engineering compression stress-strain diagrams (**c**,**d**), friction coefficient diagrams (**e**), and wear rate diagrams (**f**) of aluminum alloy AlSi5 and composite materials Al-5%Ni, Al-10%Ni.

**Table 1 materials-16-04157-t001:** Results of X-ray fluorescence analysis, filaments used, and samples obtained.

Material	Elements, wt. %											
	Al	Mn	Fe	Zn	Cr	Co	Mo	Ti	Si	Zr	Ni	V
ER4043 (AlSi5)	bal.	0.032	0.09	0.07	-	-	-	0.01	5.1	-	-	0.021
Udimet-500	1.6	0.1	0.4	0.04	17.3	13.2	4.3	2.7	0.3	0.04	bal.	-
Al–5% Ni	82.18	-	0.127	-	1.88	1.33	-	0.324	7.36	-	6.31	0.035
Al-10%Ni	66.53	-	1.08	-	5.16	3.44	-	0.654	5.35±	-	16.29	0.06

**Table 2 materials-16-04157-t002:** The chemical composition of the Al-5%Ni composite in various areas shown in Figure 8.

Spectrum	Chemical Composition, at.%						Possible Phase
	Al	Si	Ti	Cr	Co	Ni	
1	54.5	1.3	1.0	4.6	6.7	31.9	AlNi
2	63.5	5.4	1.0	4.6	5.0	20.5	Al_3_Ni
3	75.8	5.1	1.0	6.6	3.2	8.2	Al_3_(Ni,Cr,Si)
4	82.8	2.4	0.3	1.4	6.4	6.8	Al(Co,Ni)
5	97.9	1.2	0.4	0.3	0.1	0.2	α-Al
6	74.8	6.1	0.1	0.4	0.5	18.0	Al_3_Ni
7	98.8	0.7	0.2	0.1	-	0.1	α-Al
8	81.3	4.7	0.1	0.1	4.3	9.6	Al(Co,Ni, Si)
9	80.5	1.7	0.1	0.2	5.2	12.2	Al(Co,Ni, Si)
10	78.4	1.2	0.1	-	0.6	19.8	Al_3_Ni
11	80.6	3.6	0.1	-	0.5	15.3	Al_3_Ni
12	84.9	1.2	0.1	0.1	0.3	13.4	Al_3_Ni
13	72.0	18.3	0.1	-	0.3	9.4	α-Al, β-Si
14	73.0	18.7	0.1	0.1	0.2	7.8	α-Al, β-Si

**Table 3 materials-16-04157-t003:** The chemical composition of the Al-5%Ni samples in the local areas shown in Figure 9.

Spectrum	Chemical Composition, at.%								Possible Phase
	Al	Si	Ti	Cr	Co	Ni	Fe	Mo	
1	77.8	5.6	0.6	7.5	1.9	5.2	0.1	0.3	Al_3_(Ni,Cr,Si)
2	80.8	1.6	0.03	1.6	6.7	8.9	0.1	0.1	Al(Co,Ni)
3	2.0	98.0	-	-	-	-	-	-	β-Si
4	99.0	0.1	-	-	-	-	-	-	α-Al
5	1.7	98.3	-	-	-	-	-	-	β-Si
6	3.1	96.9	-	-	-	-	-	-	β-Si
7	76.1	7.2	0.2	7.3	1.9	6.3	-	-	Al_3_(Ni,Cr,Si)
8	77.3	5.5	0.3	7.6	2.1	5.6	-	-	Al_3_(Ni,Cr,Si)
9	99.9	0.1	-	-	-	-	-	-	α-Al

**Table 4 materials-16-04157-t004:** The chemical composition of the Al-10%Ni sample in the local areas shown in Figure 11.

Spectrum	Chemical Composition, at.%							Possible Phase
	Al	Si	Ti	Cr	Co	Ni	Mo	
1	42.8	1.0	1.8	7.3	7.9	39.2	-	AlNi(Cr,Co)
2	40.0	0.4	1.8	6.2	8.0	43.7	-	AlNi(Cr,Co)
3	26.9	6.0	7.0	39.2	5.1	11.6	4.3	AlCCr_2_
4	53.5	4.0	2.9	10.2	5.2	23.4	0.8	Al_3_Ni
5	73.1	4.7	0.5	9.6	3.0	8.1	1.0	Al_3_(Ni,Cr,Si)
6	73.7	5.4	0.5	9.9	2.8	6.6	1.1	Al_3_(Ni,Cr,Si)
7	72.6	0.7	0.2	0.5	4.0	22.0	-	Al_3_Ni
8	74.0	6.2	0.4	9.4	2.3	6.7	0.9	Al_3_(Ni,Cr,Si)
9	73.6	0.6	0.1	0.2	3.4	22.1	-	Al_3_Ni
10	74.0	0.5	0.2	0.3	2.6	22.4	-	Al_3_Ni
11	75.4	4.5	1.0	9.4	2.8	5.9	1.0	Al_3_(Ni,Cr,Si)
12	75.9	5.5	0.2	8.8	2.3	6.4	1.0	Al_3_(Ni,Cr,Si)
13	32.5	58.8	0.3	2.2	0.8	5.3	-	β-Si
14	96.7	2.6	0.2	0.1	0.1	0.3	-	α-Al
15	96.4	1.5	0.3	0.8	0.2	0.8	-	α-Al

**Table 5 materials-16-04157-t005:** Tensile strength values for uniaxial tension and compression in the horizontal and vertical directions, MPa.

Material	Test	Al		Al-Ni5%		Al-Ni10%	
		Top	Bottom	Top	Bottom	Top	Bottom
Vertical	Tension	142	143	74	121	57	123
Horizontal		145	164	79	59	55	88
Vertical	Compression	-	-	530	570	1150	1200
Horizontal		-	-	490	500	905	945

## Data Availability

Data sharing is not applicable to this article.

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
