# Peer review of "Al–Al3Ni In Situ Composite Formation by Wire-Feed Electron-Beam Additive Manufacturing"

_materials, 2023, doi:10.3390/ma16114157_

Round 1

Reviewer 1 Report

This article investigates on microstructure-properties relation of multimetallic composite materials prepared by wire-feed electron-beam additive manufacturing. The authors discussed the precipitation process of intermetallic compounds, and the mechanical properties were also explored on the tensile, compressive, and wear tests. However, this article is poorly organized and lacks novel results. From the innovative and objective points of view, this article is hard to reach the standard of Materials. As a reviewer, I am sorry to recommend rejecting your paper, but I would like to give you some comments as follows:

1. In the Introduction part, the author introduced the importance of Al alloys and the methods of manufacturing composite materials. But the importance of composite materials was ignored.

2. In the Experimental procedure part, the author needs to provide more details of testing conditions, such as the strain rate of the tensile and compressive experiments, as well as the size and shape of these specimens.

3. The results and discussion part should be separated into the results part and discussion part respectively.

4. The quality of Figs. 2-4 is very poor, especially Figs. 3-4 look very dirty. The corresponding analysis on the microstructure evolution lacks rigor. Usually, it is very difficult to distinguish different phases only based on the optical microstructure. Besides, the evolution in the volume fraction of the phase lacks quantitative analysis.

5. The effect of microstructure on the mechanical properties and wear resistance needs to be discussed deeply.

6. The English need to be polished by a native speaker.

Author Response

Dear Ms./Mr. Reviewer

Thank you very much for you comments

Below and in the attached file are the answers to your comments.

Best regards,

Authors team

  1. In the Introduction part, the author introduced the importance of Al alloys and the methods of manufacturing composite materials. But the importance of composite materials was ignored.

A: Added

  1. In the Experimental procedure part, the author needs to provide more details of testing conditions, such as the strain rate of the tensile and compressive experiments, as well as the size and shape of these specimens.

A: Added

  1. The results and discussion part should be separated into the results part and discussion part respectively.

A: Changed

  1. The quality of Figs. 2-4 is very poor, especially Figs. 3-4 look very dirty. The corresponding analysis on the microstructure evolution lacks rigor. Usually, it is very difficult to distinguish different phases only based on the optical microstructure. Besides, the evolution in the volume fraction of the phase lacks quantitative analysis.

A: Sorry, but we believe that the quality of the photographs is quite normal, there is no dirt on them, there is a different phase contrast, including dark areas due to the resulting porosity. Information about this has been added to the text of the article.

  1. The effect of microstructure on the mechanical properties and wear resistance needs to be discussed deeply.

A: Added

  1. The English need to be polished by a native speaker.

A: Changed

Reviewer 2 Report

Comments to the Author

The study aims to study the microstructure of composites obtained by additive electron beam technology on the basis of aluminum alloy ER4043 and Ni based Udimet500 alloy. The comments suggested are as below:

1. The title is too long.

2. The tensile properties of these composites is poor as compared to commercial materials. What is the potential application for the composites of ER4043 and Udimet500? It makes me confuse for why needs to study the phase structure of the composites.

3. The selected diffraction patterns in Fig.7 is not clear, especially the phases belong to different patterns should be listed, as well as the zone axis. There is two same (111) spots in Fig.9, which is obviously incorrect.

4. In Table.2-4, how to confirm the phase structure in various areas barely from the EDS spectrum?

5. Please explain the formation mechanism of these intermetallic phases from thermodynamic diagram calculation.

6. The expression of some terms is not correct in material science, such as “organization” needs to be corrected as “microstructure”, “multimetallic” needs to be corrected as “multiphase”, etc.

Author Response

Dear Ms./Mr. Reviewer

Thank you very much for you comments

Below and in the attached file are the answers to your comments.

Best regards,

Authors team

  1. The title is too long.

A: Changed

  1. The tensile properties of these composites is poor as compared to commercial materials. What is the potential application for the composites of ER4043 and Udimet500? It makes me confuse for why needs to study the phase structure of the composites.

A: Да, прочные свойства при растяжении получились невысокие, с этим фактом часто сталкиваются исследователи подобных композитов. однако повысились свойства при сжатии и повысилась износостойкость и соответственно есть своя ниша практического использования у таких материалов.

  1. The selected diffraction patterns in Fig.7 is not clear, especially the phases belong to different patterns should be listed, as well as the zone axis. There is two same (111) spots in Fig.9, which is obviously incorrect.

A: Fig. 9 and Fig.7 was changed.

  1. In Table.2-4, how to confirm the phase structure in various areas barely from the EDS spectrum?

A: Added to tables Table.2-4 “Possible phase”

  1. Please explain the formation mechanism of these intermetallic phases from thermodynamic diagram calculation.

A: Such information is added to the Discussion section

  1. The expression of some terms is not correct in material science, such as “organization” needs to be corrected as “microstructure”, “multimetallic” needs to be corrected as “multiphase”, etc.

A: Changed

Reviewer 3 Report

After reviewing the manuscript I would like to address the following comments/suggestions, which should be taken into consideration by the authors:

 - Line 27: Abstract: manufacturing instead of manufacturin.

- Line 32-33: “Aluminum-silicon alloys (silumins) are used to 32 produce body parts” – what body parts? That is obviously understatement. Should be corrected.

- Figure 14: it should be indicated both on the axes of the drawings and in the text describing the results of the presented results of the mechanical tests, whether it is true stress-true strain values or not.

A general comment: the purpose of this research is not clearly demonstrated. What new information this research was supposed to bring and what practical application it can have as compared to the existing solutions in this topic. The authors should clarify this in the text of the manuscript.

Author Response

Dear Ms./Mr. Reviewer

Thank you very much for you comments

Below and in the attached file are the answers to your comments.

Best regards,

Authors tea

 - Line 27: Abstract: manufacturing instead of manufacturin.

A: Changed

- Line 32-33: “Aluminum-silicon alloys (silumins) are used to 32 produce body parts” – what body parts? That is obviously understatement. Should be corrected.

A: Changed

- Figure 14: it should be indicated both on the axes of the drawings and in the text describing the results of the presented results of the mechanical tests, whether it is true stress-true strain values or not.

A: Added

-  A general comment: the purpose of this research is not clearly demonstrated. What new information this research was supposed to bring and what practical application it can have as compared to the existing solutions in this topic. The authors should clarify this in the text of the manuscript.

A: Added
